# Biogenic, anthropogenic, and sea salt sulfate size-segregated aerosols in the Arctic summer

**Roghayeh Ghahremaninezhad[1], Ann-Lise Norman[1], Jonathan P. D. Abbatt[2], Maurice Levasseur[3], Jennie L. Thomas[4]**

[1] Depatrment of Physics and Astronomy, University of Calgary, Calgary, Canada

[2] Department of Chemistry, University of Toronto, Toronto, Canada

[3] Department of Biology, Laval University, Quebec, Canada

[4] Sorbonne Universités, UPMC Univ. Paris 06, Universite Versailles St-Quentin, CNRS/INSU, UMR8190, LATMOS-IPSL, Paris, France

Correspondence to: Ann-Lise Norman (alnorman@ucalgary.ca)

## Abstract

Size-segregated aerosol sulfate concentrations were measured on board the Canadian Coast Guard Ship (CCGS) Amundsen in the Arctic during July 2014. The objective of this study was to utilize the isotopic composition of sulfate to address the contribution of anthropogenic and biogenic sources of aerosols to the growth of the different aerosol size fractions in the Arctic atmosphere. Non-sea salt sulfate is divided into biogenic and anthropogenic sulfate using stable isotope apportionment techniques. A considerable amount of the average sulfate concentration in the fine aerosols with diameter <0.49 µm was from biogenic sources (>63%) which is higher than previous Arctic studies measuring above the ocean during fall (<15%) (Rempillo et al., 2011) and total aerosol sulfate at higher latitudes at Alert in summer (>30%) (Norman et al., 1999). The anthropogenic sulfate concentration was less than biogenic sulfate, with potential sources being long range transport and, more locally, the Amundsen's emissions. Despite attempts to minimize the influence of ship stack emissions, evidence from larger-sized particles demonstrates a contribution from local pollution.

A comparison of $\delta^{34}S$ values for $SO_2$ and fine aerosols was used to show that gas-to-particle conversion likely occurred during most sampling periods. $\delta^{34}S$ values for $SO_2$ and fine aerosols were similar suggesting the same source for $SO_2$ and aerosol sulfate, except for two

samples with a relatively high anthropogenic fraction in particles <0.49 µm in diameter (July 15-17 and 17-19). The high biogenic fraction of sulfate fine aerosol and similar isotope ratio values of these particles and $SO_2$ emphasize the role of marine organisims (e.g. phytoplankton, algea, bacteria) in the formation of fine particles above the Arctic Ocean during the productive summer months.

## 1   Introduction

Climate is changing in the Arctic faster than at lower latitudes (IPCC, 2013) and it has the potential to influence the Arctic Ocean and aerosols that form above it. The Arctic ocean is considered a source of primary aerosol, such as sea salt and organic, as well as secondary particles from oxidation of $SO_2$ to sulfate ($SO_4^{2-}$) (Bates et al., 1987; Charlson et al., 1987; Andreae, 1990; Yin et al., 1990; Leck and Bigg, 2005a; Leck and Bigg, 2005b; Barnes et al., 2006; Ayers and Cainey, 2007). Aerosols drive significant radiative forcing and influence climate directly (by scattering of short/long wave radiation) and indirectly (by changing number and size of cloud droplets and altering precipitation efficiency) (Shindell, 2007). Recently, it has been shown that their net effect is cooling the Arctic which offsets around 60% of the warming effect of greenhouse gases (Najafi, et al., 2015). However, there are key uncertainties in the estimation of aerosol effects and their sources which arise from limited information on their spatial and temporal distribution.

Sulfate in the Arctic atmosphere originates from anthropogenic, sea salt and biogenic sources. Anthropogenic aerosols, with a winter-to-springtime maximum known as Arctic Haze, contain particulate organic matter, nitrate, sulfate, and black carbon which originate from North America and Eurasia (Sirois and Barrie, 1999; Quinn et al., 2002; Stone et al., 2014). Sea salt enters the atmosphere via mechanical processes such as sea spray and bubble bursting (Leck and Bigg, 2005a). Formation of breaking waves on the ocean surface (at wind speeds higher than 5 m/s) leads to the entrainment of air as bubbles into surface ocean water. These bubbles rise to the surface due to their buoyancy and start to scavenge organic matter. They burst at the air-sea interface and release sea spray aerosol (SSA) which includes organic matter and inorganic sea salt (Quinn et al., 2015). Although, sea salt is generally found in coarse mode particles, it is sometimes found in smaller sizes as well (Bates et al., 2006). Several mechanisms are responsible to formation of SSA with different sizes. Small film drops are generated by the shattering of the film caps. Larger jet drops (with size range of 1 to

25 µm) are formed by collapse of the bubble cavity. Spume drops are torn from the crests of
wave and entered directly to the atmosphere at high wind speeds, above 10 m/s (Lewis and
Schwartz, 2004; Quinn et al., 2015).
The most important source of biogenic sulfate aerosols in the Arctic summer is the oxidation
of dimethyl sulfide (DMS) (Norman et al., 1999). DMS is mostly produced by the breakdown
of its algal precursor dimethylsulfonopropionate (DMSP) by phytoplankton and bacteria
DMSP-lyases and transported from the ocean to the atmosphere via turbulence and diffusion
which depends on sea surface temperature, salinity and wind speed (Nightingale et al., 2000).
Gaseous sulfur compounds from DMS oxidation are able to form new particles or condense
on pre-existing aerosol in the atmosphere and thereby become large enough to act as Cloud
Condensation Nuclei (CCN) (Charlson et al., 1987). However, there are crucial uncertainties
in the details of the potential impact of DMS on climate at a global scale (Quinn and Bates,

13   2011).

The formation of new particles and CCN is particularly important during the summer when
anthropogenic aerosols are scarce, scavenging is efficient, and sea-atmosphere gas exchange
produces considerable DMS in the Arctic (Gabric et al., 2005; Elliot et al., 2012, Li et al.
1994; Leaitch et al., 2013). Some studies suggested an increase of biological activity, DMS
production and emission with an increase of temperature and decrease of sea-ice cover during
summer (Sharma et al. 2012; Levasseur, 2013). However, modelling results from Browse et
al. (2014) suggest that increased DMS emissions during summertime will not cause a strong
climate feedback due to the efficient removal processes for aerosol particles. Such results are
highly dependent on aerosol size distributions which are relatively unconstrained particularly
with respect to DMS oxidation (Bigg and Leck, 2001, Matrai et al., 2008; Quinn et al., 2009;
Leaitch et al., 2013).
Tracers have been used in some studies to indicate different sources for sulfate, such as the
use of DMS and MSA for biogenic activities (Savoie et al., 2002). Other studies assumed that
non-sea salt sulfur originates from biogenic sources in clean areas with low anthropogenic
sulfur emissions (Bates et al., 1992; Hewitt and Davison, 1997). These methods may
overestimate the role of biogenic sources if anthropogenic sulfate is present. The isotopic
differences of various sources present a way to determine the oceanic DMS contribution to
aerosol growth (Norman et al., 1999, 2004; Seguin et al., 2010, 2011; Rempillo et al., 2011).
Size-segregated aerosols were collected in July 2014 during an extended transect going from
the strait of Belle-Isle to Lancaster Sound in the Canadian Arctic, permitting comparison with
measurements from other seasons. Sulfate aerosols have been apportioned into biogenic,
anthropogenic and sea salt sulfate using sulfur isotopes, to find the contribution of each
source in aerosol formation and growth.

## 6    2    Field description and methods

Particles were collected on board the Canadian Coast Guard Ship (CCGS) Amundsen in the
Arctic during July 2014 as part of the NETCARE (Network on Climate and Aerosols:
Addressing Key Uncertainties in Remote Canadian Environments) project. The route of this
expedition and sampling intervals are shown in Figure 1 which took place from 8 to 24 July

11    2014.

Wind speed, and sea surface and air temperatures were documented each minute and averaged
over 10 minutes using the Automatic Voluntary Observing Ships System (AVOS) system
available onboard the Amundsen at ~23 m above the sea surface. In addition, a version of the
Lagrangian particle model, FLEXPART-WRF (Brioude et al., 2013), was used to estimate
potential emission sensitivities. More details/figures of FLEXPART-WRF are published in
other studies from the same campaign (NETCARE 2014) (e.g. Mungall et al. 2015;
Wentworth et al., 2016).
A high volume sampler was used to collect aerosol samples at a calibrated flow rate of
$1.08\pm0.05$ m$^3$/min. This high volume sampler was placed facing the bow above the bridge of
the ship, around 30 m above the sea surface. It was fitted with a cascade impactor to collect
size-fractionated particles on quartz filters as well as $SO_2$. The $SO_2$ was trapped on a cellulose
filter pre-treated with potassium carbonate ($K_2CO_3$) and glycerol solution (Saltzman et al.,
1983; Norman et al., 2004; Seguin et al., 2010). The sampling interval was two days, starting
from 10:00 h. The high volume sampler was turned off manually to avoid contamination
when the ship emissions toward the sampler were observed or at times when the ship was
stationary. Periods greater than 30 min are reported in table 1. Figure 1 shows sampling
intervals: the high volume sampler was off because of stormy weather from 10:00 h on July
19[th] to10:00 h on July 20[th]. Particle size cut off at the flow rate of 1.13 m$^3$/min and standard
temperature and pressure ($25^0$C and 1 atm) for spherical particles is at 50% collection
efficiency, and the 6 ranges of particle aerodynamic diameter of the cascade impactor are: A
(>7.2 µm), B (3.0–7.2 µm), C (1.5–3.0 µm), D (0.95–1.5 µm), E (0.49–0.95 µm), and F
(<0.49 µm). Temperature and pressurer effects are negligible, however the lower flow rate
increases slightly the cut off diameter for each size range (Tisch Environmental, 2004).
TOTAL sulfate refers to the sum of sulfate on each of the size fractions. Field blanks were
collected on two separate occasions, and loaded and unloaded with the same method as
samples processed except the high volume sampler was turned off, to assess whether and how
much contamination occurred from procedural handling and analyses. Filters were stored in
sealed ziplock bags at < 4 $^0$C before analysis in the lab.
A Li-Cor 7000 $CO_2/H_2O$ Analyzer, with an inlet near the location of the high volume sampler
(~ 3 m) and at the same height was used to measure the atmospheric $CO_2$ mixing ratios. The
objective of the $CO_2$ measurement was to determine the influence of smoke stack emissions
from the ship for QA/QC of aerosol samples. The $CO_2$ concentrations are shown in Figure 2a.
There were two periods when $CO_2$ measurements were not saved due to a computer
malfunction: 10:30 h on July 10[th] to 9:00 h on July 11[th], and 14:00 h on July 15[th] to 10:35 h on
July 17[th].  The observation shows a relatively constant $CO_2$ mixing ratio with some peaks,
indicating relatively little smoke stack contamination.
Once back in the laboratory, sulfate extracted from filters extracts was analysed for sulfate
isotopes and concentration. Filter papers were shredded in distilled deionized water and
sonicated for 30 minutes. Then, filter paper fibers were removed by 0.45 mm Millipore
filtration, and a portion of the filtrate samples (2×10 mL) was used for ion concentration
measurements. Remaining filtrate was treated with 5 mL of 10% $BaCl_2$ and 1 mL HCl to
precipitate $BaSO_4$. In addition of $BaCl_2$ and HCl, 2 mL of 30% hydrogen peroxide was added
to $SO_2$ filter solutions to oxidize the $SO_2$ to sulfate. After extraction, $BaSO_4$ was dried and
samples were packed into tin cups and analyzed with a PRISM II continuous flow isotope
ratio mass spectrometer (CF-IRMS) to obtain $\delta^{34}S$ values in parts per thousand (‰) (relative
to VCDT, Vienna Cañon Diablo Triolite) (Seguin et al., 2007). $\delta^{34}S$ for sulphur isotopes is
shown by the abundance ratio of the two principal sulfur isotopes ($^{34}S/^{32}S$) (Krouse et al.,

28   1991).

$\delta^{34}S$ (‰) = {($^{34}S/^{32}S$)sample/($^{34}S/^{32}S$)standard - 1}×1000              (1)
The uncertainty for $\delta^{34}S$ values (±0.3 ‰) was determined by the standard deviation of the
$\delta^{34}S$ values of a suite of internal standards bracketing the $\delta^{34}S$ values of the samples.
Concentrations of cations ($Ca^{2+}$, $K^+$, $Na^+$, $Mg^{2+}$) and anions ($Cl^-$, $SO_4^{2-}$, $PO_4^{3-}$, $NO_3^-$) were
obtained by ion chromatography with a detection limit of 0.1 mg/L. No peaks were detected
for sulfate in the blank filters, and the average concentration of $Na^+$ in the blank filters was
1.2 mg/L after extraction (which is around 5% and 20% of the maximum and minimum of the
$Na^+$ concentration in filter A with the most sea salt).
Three different sources - anthropogenic, biogenic, and sea salt - were considered for sulfur
aerosols and the fraction of each source was obtained using:
$[SO_4^{2-}]_{total} = [SO_4^{2-}]_{bio} + [SO_4^{2-}]_{anthro} + [SO_4^{2-}]_{SS},$        (2)
$[SO_4^{2-}]_{total}\delta^{34}S_{total} = [SO_4^{2-}]_{bio}\delta^{34}S_{bio} + [SO_4^{2-}]_{anthro}\delta^{34}S_{anthro} + [SO_4^{2-}]_{SS}\delta^{34}S_{SS}.$    (3)
Also $\delta^{34}S_{NSS}$ was determined using the expression for two source mixing:
$[NSS]\delta^{34}S_{NSS} = [measured]\delta^{34}S_{measured} - [SS]\delta^{34}S_{SS},$        (4)
where SS and NSS refer to sea salt and non-sea salt sulfate respectively, and quantities in
brackets, [X], indicate concentrations.
The amount of sea salt sulfate in sea water was calculated by $SO_4^{2-}$ and $Na^+$ mass ratios:
$[SO_4^{2-}]_{SS}=0.252[Na^+].$        (5)
Sulfur isotope apportionment in the Arctic assumes a $\delta^{34}S$ value of +21‰±0.1 (Rees et al.,
1978), +18.6‰ ±0.9 (Sanusi et al. 2006; Patris et al. 2002), and +3‰ ± 3 (Li and Barrie,
1993; Nriagu and Coker, 1978; Norman et al., 1999) for sea salt, biogenic and anthropogenic
$\delta^{34}S$ values respectively. These values were used to find sea salt, biogenic, and anthropogenic
fractions in this study. The partial derivative rule for error propagation and standard deviation
were considered for uncertainties.
**3   Results**
**3.1   The meteorological measurements**
Interaction of wind at the ocean's surface may lead to formation of primary course mode sea
salt particles. DMS oxidation pathways, the formation of biogenic $SO_2$, and production of
new particles, are influenced by wind speed and temperature. Wind speed and sea/air
temperatures from the Amundsen's AVOS system are shown in Figure 2b and 2c.

## 3.2  Sulfate aerosols

Total, sea salt, and non-sea salt sulfate concentrations and their standard deviations for the entire sampling program for different size fractions are summarized in Table 2.

Similar average sulfate concentrations were found for aerosols in $A_{>7.2\ \mu m}$ (113 ng/m$^3$), $B_{3.0–7.2\ \mu m}$ (100 ng/m$^3$), and $D_{0.95–1.5\ \mu m}$ (110 ng/m$^3$) size fractions. An average sulfate concentration of 34 ng/m$^3$ was found for the $C_{1.5–3.0\ \mu m}$ size aerosols. On the other hand, $F_{<0.49\ \mu m}$ filter (fine aerosol) has the highest average sulfate concentration (~214 ng/m$^3$) and contains less than 3% sea salt sulfate (6 ng/m$^3$).

### 3.2.1  Sea salt sulfate

Table 2 includes average sea salt sulfate concentrations for aerosols for different size fractions for this study. As expected, coarse size filters $A_{>7.2\ \mu m}$ and $B_{3.0–7.2}$ in this study contain more sea salt sulfate than smaller diameter aerosols and the average sea salt sulfate is approximately six times higher than non-sea salt sulfate. In contrast, smaller aerosols on the $D_{0.95–1.5\ \mu m}$ filter contain lower but significant amounts of sea salt sulfate (~ 55 ng/m$^3$). Although, on average, more than 75 percent of sulfate for the $C_{1.5–3.0\ \mu m}$ filter is from sea salt, a considerable decrease in concentration is observed compared to $A_{>7.2\ \mu m}$, $B_{3.0–7.2\ \mu m}$ and $D_{0.95–1.5\ \mu m}$ filters. Sea salt sulfate concentrations are low for aerosols collected on the $E_{0.49–0.95\ \mu m}$ and $F_{<0.49\ \mu m}$ filters (~ 5 to 6 ng/m$^3$). The spatial variability of TOTAL sulfate and sea salt concentrations is shown in Figure 3a.

### 3.2.2  Non-sea salt sulfate

The average non-sea salt sulfate concentrations for the entire study are reported in Table 2 (spatial variation in non-sea salt sulfate is shown in Figure 3b). Results show approximately uniform TOTAL non-sea salt sulfate concentrations (average 130±21 ng/m$^3$: range from 102 to 152 ng/m$^3$), except the first sample collected nearby the Gulf of St Lawrence (July 8[th] to 10[th]) which contains the highest non-sea salt sulfate concentration. The majority of sulfate for small aerosols in the $D_{0.95–1.5\ \mu m}$ (~ 55 ng/m$^3$, 50%), $E_{0.49–0.95\ \mu m}$ (~ 66 ng/m$^3$, 93%) and $F_{<0.49\ \mu m}$ (~ 208 ng/m$^3$, 97%) fractions is from non-sea salt sources.

# 4    Discussion

## 4.1    Sea salt sulfate

Sea salt concentrations are variable with season and depend on atmospheric stability (Lewis and Schwartz 2004). Although wind is considered as an important factor to sea-air exchange of sea salt, correlations in this study between wind speed and sea salt sulfate concentrations for coarse and fine mode aerosols were not significant ($R^2 \cong 0.1$), which is consistent with previous studies (Lewis and Schwartz 2004; Rempillo et al., 2011; Seguin et al., 2011; Jaeglé et al. 2011).

## 4.2    Non-sea salt sulfate

The spatial variation of non-sea salt sulfate (anthropogenic plus biogenic aerosols) is shown in Figure 3b. Results show approximately uniform non-sea salt sulfate concentrations for samples in the Labrador Sea and north ($130\pm21$ ng/m$^3$). Sulfate concentrations, especially non-sea salt sulfate, in this research were found to be higher than previous Arctic studies above the ocean during fall (2007-2008) (Rempillo et al., 2011), at higher latitudes at Alert in summer (1993-1994) (Norman et al., 1999) and about the same as at Barrow, Alaska during July (1997-2008) (Quinn et al., 2009). One reason could be higher biological activity and biogenic aerosols from phytoplankton during summer as addressed in the next section.

## 4.3    Sulfur isotope apportionment

Total $\delta^{34}$S (Equation 2) versus the percentage of sea salt sulfate of size fractionated aerosols is shown in Figure 4. The mixing lines for sea salt/biogenic sulfate (solid line) and sea salt/anthropogenic sulfate (dashed line) are shown to demonstrate mixing for each pair of sources. Data from this study fall mainly within the mixing lines which suggests the assignment of the end-member $\delta^{34}$S values is appropriate. However it can also be seen the data lie in two groups. One cluster has a high percent sea salt sulfate (>40% to >95%) and the second has a very low percent (<10%) sea salt sulfate.  There is a high contribution of sea salt sulfate for aerosols on filters A$_{>7.2\ \mu m}$ and B$_{3.0-7.2}$ and this decreases for smaller size aerosols. Sulfate aerosols on the A$_{>7.2\ \mu m}$ filter lie along the sea salt/anthropogenic mixing line and are consistent with sea spray and a small contribution from the ship's stack emission. Aerosols on the B$_{3.0-7.2\ \mu m}$, C$_{1.5-3.0\ \mu m}$ and D$_{0.95-1.5\ \mu m}$ filters and most of the E$_{0.49-0.95\ \mu m}$ filters lie between

the upper and lower mixing line near to the right hand side of the Figure 4. This indicates that sulfate is dominated by sea salt for these samples and the remainder is a mixture of biogenic and anthropogenic sulfate. The $\delta^{34}S$ value for aerosols <0.49 microns ($F_{<0.49 \ \mu m}$ filter) is more variable, it indicates very little sea salt sulfate is present and the majority of the sulfate is derived from a mixture of biogenic and anthropogenic sulfate. Norman et al. (1999) showed that most data from Alert during spring, fall, and winter lie between 0 and +7‰ which demonstrates a combination of anthropogenic and sea salt sulfate aerosols. Also, their data show an increase in $\delta^{34}S$ values during summer (between +7‰ and +15‰) and confirm the importance of biogenic sulfate. The $\delta^{34}S$ data for non-sea salt sulfate from Rempillo et al. (2011) illustrate the dominance of anthropogenic sources (more than 70%) during fall 2007 and 2008. In addition, Rempillo et al. (2011) introduced a new sulfate source, the Smoking Hills ($\delta^{34}S$ = -30‰). This new source altered background $\delta^{34}S$ to -30‰ near the Smoking Hills on Cape Bathurst, Northwest Territories (Figure 1) and $\delta^{34}S$ = -5‰ further away. There is no evidence from the isotope data for a significant contribution of sulfate from the Smoking Hills in this study, however, results from FLEXPART-WRF modeling show several potential emissions originated or passed near the Smoking Hills (Figure 5).

## 4.4  Anthropogenic and biogenic sulfate

The concentration of sulfate for aerosol samples derived from apportionment calculations for non-sea salt sulfate, anthropogenic and biogenic sources is shown in Figure 6. Results show an approximately uniform concentration ($130\pm21$ ng/m$^3$) for sulfate aerosols in the Arctic region, aside from the Gulf of the St. Lawrence which has around four times higher concentrations (Figure 6a). In addition, the highest concentration for both anthropogenic and biogenic sulfate were found in the $F_{<0.49 \ \mu m}$ filter in the Arctic region.

Two possible sources for anthropogenic sulfate are ship emissions and long range transport (LRT). In the Arctic $CO_2$ above background is likely from ship emissions. The question is what is the appropriate background $CO_2$ mixing ratio? Analyses were performed assuming three different levels for background $CO_2$ (380, 385, 400 ppm). The result of these analyses indicates that $CO_2$ mixing ratios (Figure 2a) reached 380, 385 and 400 ppm for less than 1.5, 0.5 and 0.1% of sampling time respectively and were relatively uniform in comparison with similar measurements by Rempillo et al. (2011) which reached more than 2000 ppm when stack emissions impacted the samples, on average, 5% of the sampling time (O. Rempillo,

Personal communication June 2015). Therefore, the direct impact of ship stack emissions on most aerosol samples in this study collected is expected to be small. This was confirmed by nearly white filter samples after collection for all size fractions during this study compared to filters which appeared grey or black when contaminated by ship stack sulfate in the SOLAS study from 2007 to 2008 (O. Rempillo, Personal communication June 2015; Rempillo et al., 2011). Furthermore, weak correlations were observed between anthropogenic sulfate and $CO_2$ for the $A_{>7.2\ \mu m}$, $B_{3.0-7.2\ \mu m}$, $D_{0.95-1.5\ \mu m}$, $E_{0.49-0.95\ \mu m}$, and $F_{<0.49\ \mu m}$ samples suggesting that some portion of the anthropogenic sulfate was locally derived from the ship's emissions. However, the correlations were poor so $CO_2$ is not considered as an adequate tracer to distinguish local sulfate from LRT.

Long range transport of $SO_2$ and particles is a second potential mechanism affecting the concentration of anthropogenic sulfate during this study. The lifetime for $SO_2$ in the Arctic is more than one week (Thornton et al., 1989) and this potentially acts as a reservoir from which new anthropogenic aerosols could form. Long range transport of anthropogenic sulfur dominates in the Arctic winter and early spring because of the stable atmosphere and weak removal of particles, and concentrations significantly decrease during summer because of a lower number of sources within the polar front and stronger scavenging (Quinn et al., 2002; Stone et al., 2014). The backward configuration modeling of FLEXPART-WRF shows that potential emissions originated from the east for the first few days (12[th], 13[th]), and expanded to cover a broader region after that (Figure 5 shows some examples of backward configuration results of FLEXPART-WRF). The Hudson Bay area is an important source of DMS (Richards et al., 1994), and air parcels originating from Hudson Bay may contain more biogenic $SO_2$ and sulfate. On the other hand, air parcels originating from the south (North America) may contain more pollution from LRT.

Figure 6b shows the time series of anthropogenic sulfate concentrations for size segregated aerosols. The size fraction of aerosols is different for two distinct anthropogenic sources: long range transport and ship emissions. The contribution of anthropogenic sulfate from long range transport is highest for the first sample collected in the Gulf of St. Lawrence and is pronounced in the $E_{0.49-0.95\ \mu m}$ and $F_{<0.49\ \mu m}$ filters. On the other hand, the anthropogenic aerosol sulfate concentrations on filters $A_{>7.2\ \mu m}$, $B_{3.0-7.2\ \mu m}$, and $C_{1.5-3.0\ \mu m}$ were highest for samples collected from July 17[th] to 19[th], which suggests more sulfate from the ship's emissions. Although the high volume sampler was turned off when the ship was stationary on

each of these days, some anthropogenic aerosols from ship emissions may have influenced the results for aerosol sulfate in that time period (July 17$^{th}$ to 19$^{th}$).

A considerable amount of the sulfate concentration, ranging from 18 to 625 ng/m$^3$ for F$_{<0.49 \, \mu m}$ filters, is from biogenic sources. These values are higher than previously measured in the Arctic. For example, the average biogenic TOTAL sulfate concentration at Alert was around 30 ngS/m$^3$ during July (Norman et al., 1999). Also, Rempillo et al. (2011) reported low biogenic sulfate concentrations with maximum and median equal to 115.2 and 0 ng/m$^3$ respectively, above the Arctic Ocean in the Canadian Archipelago during fall 2007 and 2008.

Figure 6b and 6c show that filter F$_{<0.49 \, \mu m}$ contains the highest biogenic and anthropogenic sulfate concentrations for all samples (except anthropogenic sulfate for July 11-13). The biogenic fraction of non-sea salt sulfate for each size range is reported in table 3: high fractions of sulfate on filter F$_{<0.49 \, \mu m}$ were from biogenic sources (73, 95, 92, 65%), except two samples collected on July 15-17 (25%) and 17-19 (41%) (see section 45).

## 4.5 Aerosol growth

The oxidation of SO$_2$ occurs in the gas phase, the aqueous phase, and also on the surface of particles. The rate of this oxidation depends on factors such as the presence of the aqueous phase in the form of clouds and fogs, the concentration of oxidants such as H$_2$O$_2$ and O$_3$, cloud pH, and sunlight intensity. The $\delta^{34}$S value of aerosols reflects the proportion of $\delta^{34}$S values for pre-existing aerosols and SO$_2$, by oxidation of local SO$_2$ on the surface of, or within, pre-existing aerosols (Seguin et al., 2011). Although the $\delta^{34}$S value for pre-existing aerosols is not clear, it is reasonable to assume that particles with different sizes and the same $\delta^{34}$S value originate from the same source (Seguin et al., 2011). However, sulfur isotope fractionation can confound apportionment. Harris et al., (2013) reported sulphur isotope fractionation due to SO$_2$ oxidation, which depends on temperature and oxidation pathways. By solving isotope fractionation equations (Harris et al., 2013) for the average temperature during sampling for this study (~5°C), $\delta^{34}$S values of sulfate are (10.6 $\pm$ 0.7)‰, (16.1 $\pm$ 0.1)‰, and (-6.22 $\pm$ 0.02)‰ for homogeneous, heterogeneous, and TMI oxidation, respectively. However, a comparison of the $\delta^{34}$S values for SO$_2$ and the F$_{<0.49 \, \mu m}$ filter (or any other size fractions) does not support consistent isotope fractionation during SO$_2$ oxidation for samples collected during this campaign.

The isotope ratios ($\delta^{34}$S value) for $F_{<0.49\ \mu m}$ and $SO_2$ filters are shown in Figure 7 along with
the 1:1 line. Four of six samples lay close to the 1:1 line which suggests they have the same
source or mixture of sources (and same isotope ratio value). However, there are two samples,
collected on July 15-17 and 17-19, with different $\delta^{34}$S values for $SO_2$ and $F_{<0.49\ \mu m}$ filter
sulfate, which are shown with an asterisk on Figure 7. The anthropogenic fraction of sulfate
for the $F_{<0.49\ \mu m}$ filter for these two sampling periods is relatively high. Although, the
anthropogenic fraction of sulfate in $F_{<0.49\ \mu m}$ filters for these two sampling periods was higher
than the remainder of samples (refer to section 4.4), $SO_2$ was predominantly biogenic (more
than 80%).
Conditions for aerosol nucleation based on biogenic $SO_2$ concentratrions were evaluated by
Rempillo et al. (2011). They showed that the threshold value of biogenic $SO_2$ to form new
particles was 11 nmol/m$^3$ for the clean Arctic atmosphere in fall. Sulfur dioxide
concentrations in this study were higher than this threshold throughout the July 2014
campaign (average around 32 nmol/m$^3$) except for July 11-13. This is consistent with the
measurements of Mungall et al. (2015) who reported high DMS concentrations in both the
ocean and atmosphere during the same cruise. When $\delta^{34}$S values for aerosol size fractions and
$SO_2$ are similar, then it is likely that local $SO_2$ oxidation lead to substantial sulfate content.
There are two periods that this is clearly the case and biogenic sulfate was dominant:

19         1. July 11-13 with $\delta^{34}$S values for $E_{0.49-0.95\ \mu m}$ and $D_{0.95-1.5\ \mu m}$ filters of +14.2 and +13.1

20         ‰ respectively and,

21         2. July 13-15 with $\delta^{34}$S values for $SO_2$, $F_{<0.49\ \mu m}$ and $E_{0.49-0.95\ \mu m}$ filters of +16.7, +16.8

22         and +15.8 ‰ respectively.

In contrast, anthropogenic sulfate contributed to aerosol growth on July 9-11 with $\delta^{34}$S values
for $E_{0.49-0.95\ \mu m}$ and $D_{0.95-1.5\ \mu m}$ filters equal to +5.4 and +5.0 ‰ respectively.
It is interesting to note that $\delta^{34}$S values for July 17-19 on the $E_{0.49-0.95\ \mu m}$ filters (0.49-0.95 µm)
and $SO_2$ indicate almost pure biogenic sulfur ($\delta^{34}S_E$ = +17.8 ‰, $\delta^{34}S_{SO2}$ = +17.6 ‰).
However, the $\delta^{34}$S value for sulfate on the $F_{<0.49\ \mu m}$ filters (<0.49 µm) was lower, +10.2 ‰.
This suggests aerosols <0.49 µm (F) for this sampling period originated, in part, from
anthropogenic sources, but aerosol growth from 0.49 to 0.95 µm (E) was dominated by
oxidation of biogenic $SO_2$ at this time.

## 5    Conclusion

Size segregated aerosol sulfate concentrations were measured in the Arctic and sub-Arctic during July 2014. Sulfate was apportioned between sea salt, biogenic and anthropogenic sources using sulfur isotopes. Around 85% of coarse mode (>0.95 µm) aerosol sulfate was from sea salt. However there was little to no sea salt sulfate in fine aerosols (<0.49 µm), and more than 97% of the sulfate in these aerosols was non-sea salt. Approximately uniform non-sea salt sulfate concentrations were found for TOTAL sulfate ($130\pm21$ ng/m$^3$) in the Arctic atmosphere. The dominant source for fine aerosols and $SO_2$ was biogenic sulfur, arising from oxidation of DMS, which is likely due to high ocean-atmosphere gas exchange and the large ice-free surface in the Arctic during July (Levasseur, 2013).

A comparison of $\delta^{34}S$ values for fine (<0.49 microns) aerosols and $SO_2$ samples was used to show that the growth of pre-existing fine particles occurred primarily from the oxidation of $SO_2$ from DMS during all sampling events except two where a relatively high anthropogenic fraction in the smallest submicron size (<0.49 microns, F filter) was found (July 15-17 and 17-19). The dominance of ocean biogenic sources in fine aerosol sulfate and the similarity of the sulfur isotope composition for $SO_2$ and these fine particles highlight the contribution of marine life to the formation/growth of fine particles above the Arctic Ocean during the productive month of July.

## Acknowledgements

This study was part of the NETCARE (Network on Climate and Aerosols: Addressing Key Uncertainties in Remote Canadian Environments) project and was supported by funding from NSERC. The authors also would like to thank the crew of the Amundsen and fellow scientists.

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

29

Table 1. Periods greater than 30 min when the high volume sampler was off to avoid
contamination from ship emissions. The sampling interval was two days, starting from 10:00
h.

| Sampling interval (July 2014) | Turn off-on time (UTC) of the high volume sampler | Reason to turn off the high volume sampler |
|---|---|---|
| 9-11 | July 10: 12:40 h-13:10 h | Ship emissions toward the sampler |
| 11-13 | July 11: 11:20 h-13:30 h | To change the sampler exhaust |
| 13-15 | July 15: 6:30 h-8:00 h | The ship was stationary |
| 15-17 | July 17: 8:00 h-10:00 h | The ship was stationary |
| 17-19 | July 18: 22:00 h-7:00 h[*] | The ship was stationary |
| 20-22 | July 21: 15:30 h-16:10 h | Ship emissions toward the sampler |

[*] 7:00 h on the following day July 19

1 Table 2. Average TOTAL, sea salt and non-sea salt sulfate concentrations ($ng/m^3$), sulfur

2 isotopic values (‰), and non-sea salt fraction (%) for size segregated aerosols filters.

3 Standard deviations are reported in parentheses.

| Filter Size (µm) | Average Sulfate ($ng/m^3$) | Average $\delta^{34}S$ (‰) | SS Sulfate ($ng/m^3$) | NSS Sulfate ($ng/m^3$) | Fraction of NSS Sulfate (%) |
|---|---|---|---|---|---|
| $A_{>7.20\ \mu m}$ | 113 (93) | +18.9 (1.1) | 99 (85) | 14 (13) | 12 |
| $B_{3.00–7.20\ \mu m}$ | 100 (82) | +18.2 (1.2) | 86 (75) | 14 (8) | 14 |
| $C_{1.50–3.00\ \mu m}$ | 34 (20) | +18.0 (0.6) | 27 (20) | 8 (1) | 23 |
| $D_{0.95–1.50\ \mu m}$ | 110 (200) | +16.0 (2.3) | 55 (93) | 55 (110) | 50 |
| $E_{0.49–0.95\ \mu m}$ | 71 (130) | +12.3 (5.8) | 5 (5) | 66 (120) | 92 |
| $F_{<0.49\ \mu m}$ | 214 (320) | +14.0 (1.5) | 6 (6) | 208 (320) | 97 |

Table 3. Biogenic fraction of non-sea salt sulfate (%) for each size range of filter. There was
not enough sample for isotope analysis for some periods.

| Filter Size (µm) /Sampling intervals | 09-11 | 11-13 | 13-15 | 15-17 | 17-19 | 20-22 |
|---|---|---|---|---|---|---|
| $A_{>7.20\ \mu m}$ | 42 | 44 | - | 54 | - | 14 |
| $B_{3.00–7.20\ \mu m}$ | 28 | 22 | - | 31 | - | 44 |
| $C_{1.50–3.00\ \mu m}$ | - | 51 | 47 | - | - | 45 |
| $D_{0.95–1.50\ \mu m}$ | 13 | 67 | 47 | - | - | 66 |
| $E_{0.49–0.95\ \mu m}$ | 15 | 74 | 85 | - | - | 30 |
| $F_{<0.49\ \mu m}$ | 73 | 95 | 92 | 25 | 41 | 65 |

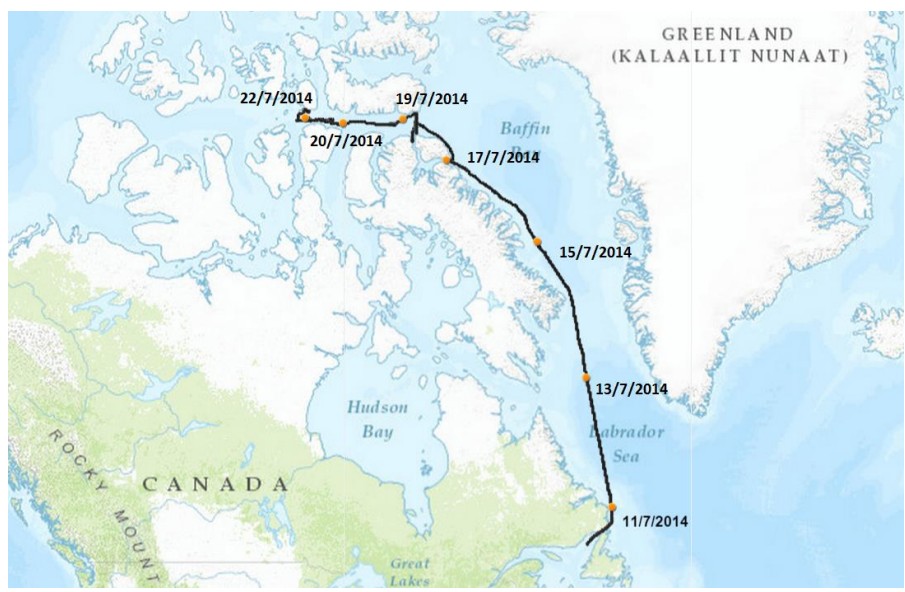

Figure 1. The route of CCGS *Amundsen* from 8 to 24 July 2014. Circles indicate sampling intervals
for the high volume sampler from 9 to 22 July (9-11, 11-13, 13-15, 15-17, 17-19, 20-22). The high
volume sampler was off because of stormy weather from 10:00 h on July 19[th] to10:00 h on July 20[th].

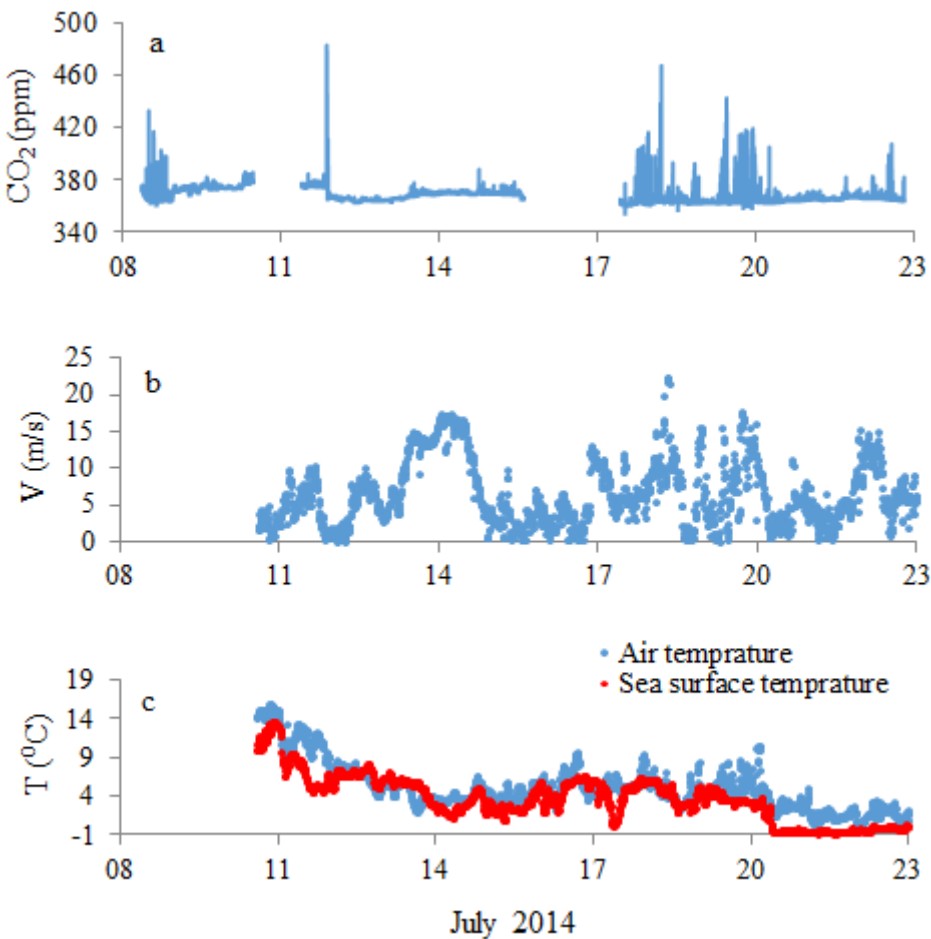

2      Figure 2. (a) $CO_2$ mixing ratio (ppm); (b) Wind speed (m/s) (c) Sea surface and Air temperatures ($^0$C).

3      $CO_2$ measurements were not reported from 10:30 h on July 10[th] to 9:00 h on July 11[th], and 14:00 h on

4      July 15[th] to 10:35 h on July 17[th]. Wind speed and tempreatures were nor recorded before July 11[th].

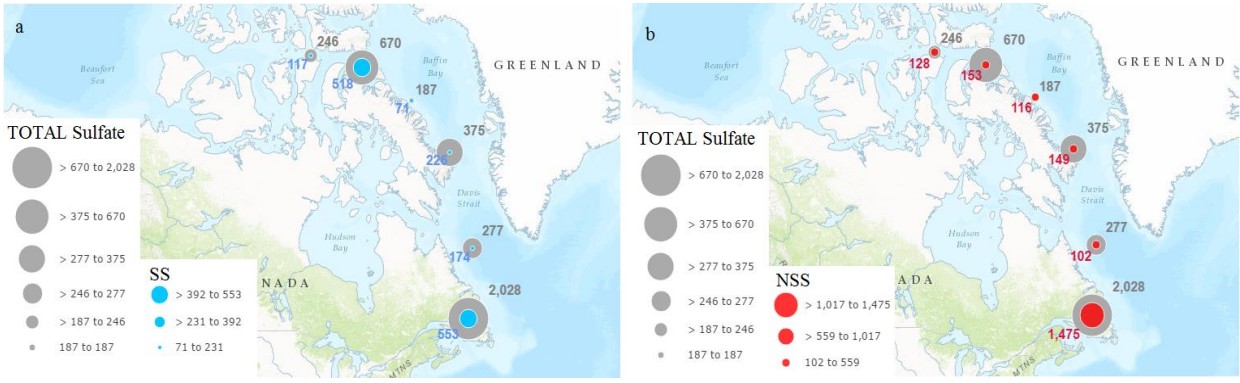

Figure 3. TOTAL sulfate, sea salt (a) and non-sea salt (b) sulfate concentrations (ng/m$^3$) of aerosols on A$_{>7.2\ \mu m}$-F$_{<0.49\ \mu m}$ filters. Numbers in the figure show TOTAL, sea salt and non-sea salt sulfate concentrations (ng/m$^3$) in gray, blue, and red colors respectively.

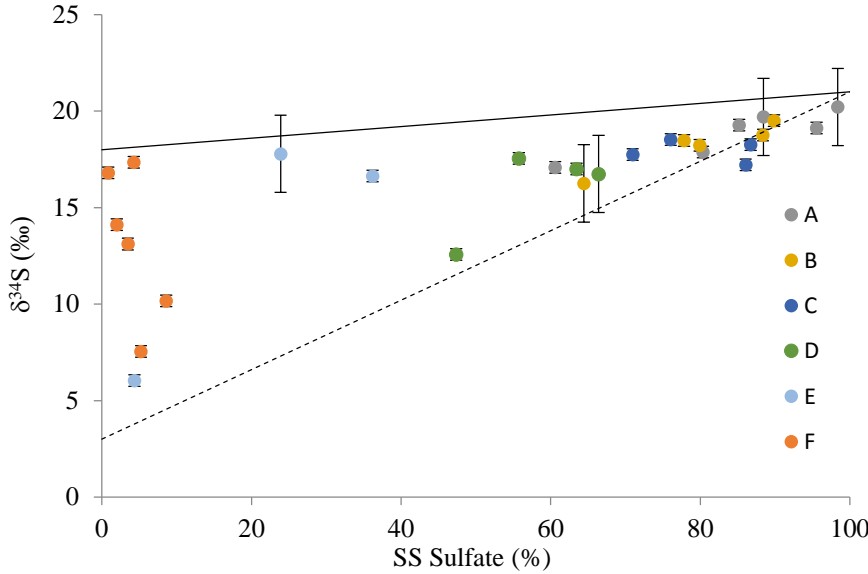

2   Figure 4. Total δ³⁴S versus the percentage of sea salt sulfate of size fractionated aerosols. The mixing

3       lines show sea salt/biogenic sulfate (solid line) and sea salt/anthropogenic sulfate (dashed line)

4       contributions. The standard deviations of each run were taken as the uncertainty for δ³⁴S values.

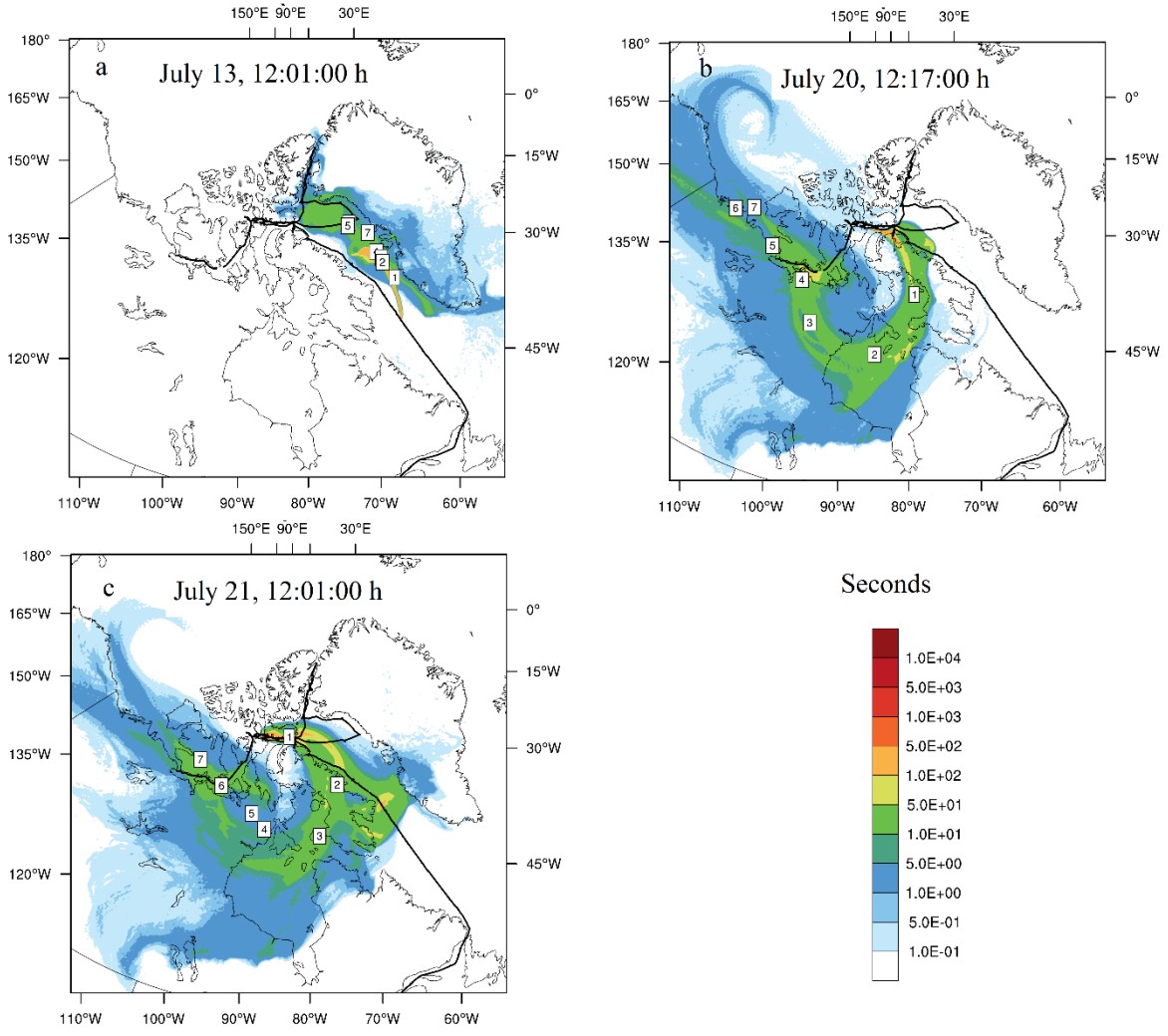

Figure 5. FLEXPART-WRF backward configuration of potential emission sensitivity plots for a) July
13 (12:01:00 h), July 20 (12:17:00 h), and July 21 (12:01:00 h). The black line shows the ship track
(note that these panels include the ship track after July 23th 2014 when high volume sampling was not
performed). The airmass residence time (seconds) before arriving at the ship location is shown with
different colors. Numbers on the panels show the approximate lifetime and the center of the plume
locations.

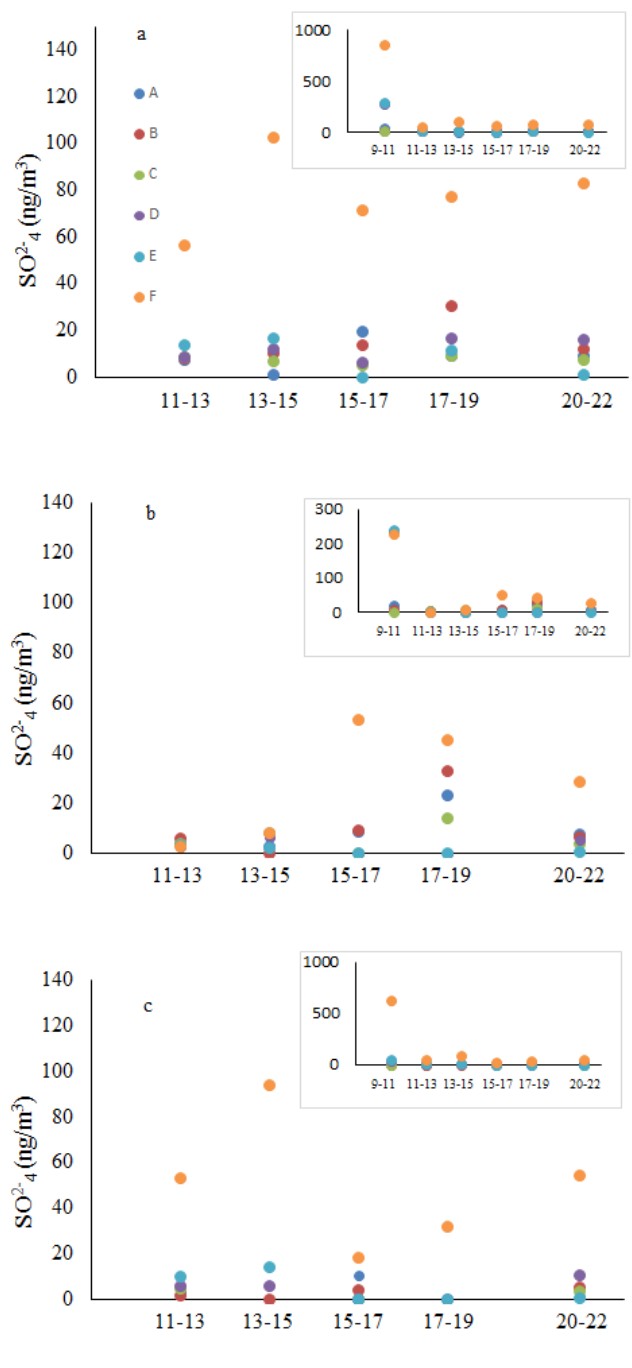

2    Figure 6. Non-sea salt (a), anthropogenic (b) and biogenic (c) of non-sea salt sulfate concentrations,

3        for size segregated aerosols in the Arctic and sub-Arctic. Strictly Arctic samples include thoes

4        collected after July 13th. Inserts contain the first sampling period (9-11 July) in the Gulf of St.

5                                        Lawrence.

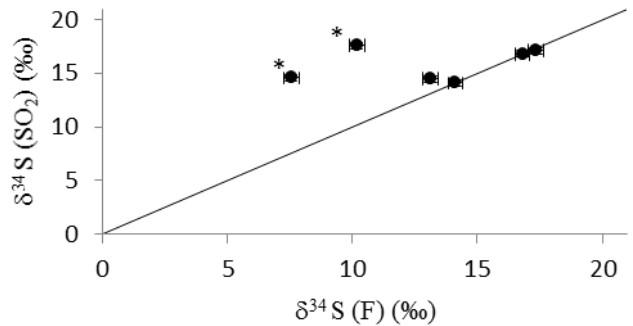

Figure 7. The isotope ratio ($\delta^{34}$S value) for F$_{<0.49\ \mu m}$ and SO$_2$ filters along with the 1:1 line. Two samples with different $\delta^{34}$S values for SO$_2$ and F$_{<0.49\ \mu m}$ filter sulfate are shown with asterisks.

