# Peer review of "Biogenic, anthropogenic, and sea salt sulfate size-segregated aerosols in the Arctic summer"

_Atmospheric Chemistry and Physics, 2015_

## Referee Comment (RC1) · Anonymous Referee #2 · 1 Mar 2016

Review of manuscript "Biogenic, anthropogenic, and sea salt sulfate size-segregated aerosols in the Arctic summer" by Ghahremaninezhad et al.

This is a fair manuscript presenting a data set of isotopic sulfates during a summer cruise in the Arctic. The study aims to use this data set to assess biogenic and anthropogenic sources and their contribution to aerosol of different size fractions. The cruise track (study area) is also interesting, as the Arctic is facing a lot of fast changes and data like this is much required. Overall, the scope of the study is relevant for ACPD/ACP and the method presented is rather scientifically sound.

General comments:

I think the paper would benefit from consistent re-structure/introduction of the samples, how many, which one is which as right now it is rather confusing moving from one figure to another. For example, one sample is called July 15-17, and then in Figure 5 you have to look for July 16 point, whereas in another figure (figure 3) you have to remember what day the samples were taken to extract relevant information, since it was not indicated at all there. Please also see the relevant detailed comments that I made in the later section.

The method and discussion part should be revised to also critically evaluate any uncertainty in the measurements, which might affect the results shown.

Specific comments:

Page 2, line 23: "Sea salt enters the atmosphere via mechanical processes such as sea spray and bubble bursting" – this sentence is ambiguous. It could be good to explain briefly how sea spray aerosol is formed, with relevant references, such as (Lewis and Schwartz, 2004; Quinn et al., 2015).

Page 4, line 12-14: "The high volume sampler was turned off manually to avoid contamination when the ship's emissions toward the sampler were observed or at times when the ship was stationary" - Can you specify how often / how long are these period?

Section 2: What are the uncertainties of the CF-IRMS?

Section 2: Please comment on the performance / uncertainty of the cascade impactor and how they might affect your results.

Section 2: So how many samples did you collect in total? If the sampling period is 16 days (8-24 July) and your sampling interval is 2 days, then did you have 8 samples? Then why in Figure 3 you seemed to have only 6 data points? Please explain.

Section 2, page 5, line 21-24: You cited some sulfur isotope apportionment in the Arctic. Did you use this in your calculations shown here? Please specify.

**ACPD**
Section 2: Please include some short description of  $\delta$ 34S

Page 8, line 15: "shows" should be "show"

Page 8, line 17: Please remove ":"

Page 9, line 1, 2: should be "ship emissions"

Page 9, line 13: the grey filters from 2007, 2008: which study was this? Was it mentioned in the study? Please cite.

Page 9, line 17: A re-definition of LTR.

Page 10, line 9: Please remove ":"

Table 1: This table display and format could be modified so that it is easier to pick out important information. There are too many brackets, e.g. Average sulfate (stdev) (ng/m3), hence confusing. Also the authors should avoid using too many horizontal and vertical lines in the table.

Figure 1: This is a fine transect.

Figure 2: The time duration of the 3 graphs are not the same. I suggest that the time duration should match the sampling interval (8-24 July?), and please specify when support data is not available.

Figure 3: I would suggest using different color codes for SS and NSS.

Also, it seems that in Figure 3a, sea salt sulfate was higher than total sulfate (second point from the top). It would be good to have detailed temporal data in number, so that it is easier to use and compare later, not just as average as currently in Table 1.

Figure 5: This figure is blurry and hard to read. Also, it should be SO42-. Please also specify which day/which samples were considered more "Arctic", as it is difficult to flip back and forth to the transect figure to find out. I would suggest to name the sample 1, 2, 3, 4, 5, 6 or something, and keep the same consistent names in relevant figures and
discussions.

References Lewis, E. R., and Schwartz, S. E.: Sea salt aerosol production : mechanisms, methods, measurements and models : a critical review, Geophysical monograph, 152, American Geophysical Union, Washington, DC, xii, 413 p. pp., 2004. Quinn, P. K., Collins, D. B., Grassian, V. H., Prather, K. A., and Bates, T. S.: Chemistry and Related Properties of Freshly Emitted Sea Spray Aerosol, Chem Rev, 115, 4383-4399, 10.1021/Cr5007139, 2015.

---

## Referee Comment (RC2) · Becky Alexander (Referee) · 24 Mar 2016

Ghahreman et al. report observations of size-segregated concentration and isotopic composition of sulfate aerosol collected in July during an Arctic cruise. The sulfate and sodium concentrations along with the sulfur isotopes are used to apportion the sea salt, marine biogenic and anthropogenic sources of sulfur in the different size bins. They find that the majority of the sulfate in the fine-mode aerosol originates from biogenic sources.

The paper is generally well written. One important thing to address is the magnitude of Na+ blanks on the filters. On page 6 they mentioned the lack of a sulfate blank, but Na+ was not mentioned. Since they use Na+ to correct for sea-salt sulfate, and seasalt sulfate is such a large fraction of total sulfate, this is critical as it strongly influences the value of their calculated d34Snss and thus their conclusions. Also, I think it would be much more informative to give the total sulfate concentration, and then the fraction of each source. Instead the authors give the absolute concentrations of each source. I can't tell how important each source is unless I go back through the text and figures to find out the total sulfate concentration. If you still want to focus on the absolute concentrations of each fraction, please also give the fractions in parentheses.

Other, more specific issues:

Page 6: What is the analytical precision of the S-isotope measurements and how was it determined?

Page 7: Wind speed also influences DMS emissions.

Page 7 and throughout the paper: There are a lot of seemingly quantitative statements in the manuscript without the numbers in the text to back them up. For example, on page 7 line 13, how much less sea salt sulfate does it contain? Page 8 line 8, "the majority of sulfate" – what fraction is "majority"? Page 9 line 10, define what you mean by "high" and "low". Page 9 line 22: what percent makes this "important"? Page 9 line 23, what percent makes this "dominant"?

Page 8 line 17: Also should cite Jaeglé et al. [2011].

Page 10: The FLEXPART-WRF model results should be presented. I was expecting to see a plot of the back trajectories but this seems to be missing.

Page 10 line 2: There is no evidence from the isotope data for a significant contribution. . .

Page 10 line 19: How were sensitivity tests performed? Are you running a model? More detail is needed here.

Page 11 line 20: Again, plots showing the results of the back trajectory calculations

would be useful to show in a figure and referred to here.

Paragraph beginning on Page 11 Line 21: Shouldn't you be discussing the biogenic contribution here? It seems weird to ignore it here when it's so important.

Page 12 line 6: replace "in solution" with "the aqueous phase".

Page 12 lines 6-8: Cloud pH also strongly influences the rate of aqueous-phase reactions.

Page 12 lines 14-19: How were these numbers calculated? You have to assume some value for d34S(SO2) which is not stated here.

Page 13 line 4: Would this make these samples biased high in the calculated anthropogenic fraction?

Page 13 line 22: from 0.49 to 0.95

Table 1: Include fraction of nss-SO4 here.

Figure 4: What do the error bars represent in Figure 4 and how were they calculated? There is no dashed line in my version of the figure.

Figure 5: I would find this figure more useful if b and c showed fractions instead of absolute concentrations.

References: Jaeglé, L., P. K. Quinn, T. S. Bates, B. Alexander, and J.-T. Lin (2011), Global distribution of sea salt aerosols: New constraints from in situ and remote sensing observations, Atmos. Chem. Phys., 11, 3137-3157, doi:doi:10.5194/acp-11-3137-2011.

---

## Author Comment (AC1) · 31 Mar 2016

Dear Reviewer,

We are very thankful for your great comments. We believe that we have addressed all of the concerns. Please find the attached supplement: the revised sentences and sections in the supplement are highlighted with yellow color.

Yours Sincerely, Roghayeh Ghahremaninezhad, (PhD Candidate) Department of Physics and Astronomy, University of Calgary, Tell: +1 403 708 2332 Email address: r.gh.phy@gmail.com rghahrem@ucalgary.ca

Comments and answers:

1. Page 2, line 23: "Sea salt enters the atmosphere via mechanical processes such as sea spray and bubble bursting" – this sentence is ambiguous. It could be good to explain briefly how sea spray aerosol is formed, with relevant references, such as (Lewis and Schwartz, 2004; Quinn et al., 2015). Supplement, page 2, line 25 to page 3 line 3: More details were added and we have now referred to "Lewis and Schwartz, 2004" and "Quinn et al., 2015".

2. Page 4, line 12-14: "The high volume sampler was turned off manually to avoid contamination when the ship's emissions toward the sampler were observed or at times when the ship was stationary" - Can you specify how often / how long are these period? Supplement, page 4, line 27: Table 1 (page 19) has been added to show periods when the high volume sampler was off for more than 30 minutes and why it was turned off.

3. Section 2: What are the uncertainties of the CF-IRMS? Supplement, page 5, line 30: The standard deviation in replicate measurements of the standards is ±0.3 ‰

4. Section 2: Please comment on the performance / uncertainty of the cascade impactor and how they might affect your results. Supplement, page 4, line 29: More details were added on the performance / uncertainty of the cascade impactor.

5. Section 2: So how many samples did you collect in total? If the sampling period is 16 days (8-24 July) and your sampling interval is 2 days, then did you have 8 samples? Then why in Figure 3 you seemed to have only 6 data points? Please explain. Supplement, page 4, line 28, and caption of figure 1 (page 22): Sampling intervals for the high volume sampler: from 9 to 22 July (9-11, 11-13, 13-15, 15-17, 17-19, 20-22). Six samples were collected. The first day (July 8) was close to Quebec City so sampling was started the next day. Also, the high volume sampler was off because of stormy weather from 10:00 h on July 19th to10:00 h on July 20th.

6. Section 2, page 5, line 21-24: You cited some sulfur isotope apportionment in the Arctic. Did you use this in your calculations shown here? Please specify. Supplement, page 6, line 19: Yes, we used the values. We added a sentence to make this clear.

7. Section 2: Please include some short description of $\delta34S$. Supplement, page 5, line 26: Description of $\delta34S$ has been added.

8. Page 8, line 15: "shows" should be "show". Supplement, page 9, line 15: Thank you, we corrected that.

9. Page 8, line 17: Please remove ":". Supplement, page 9, line 17: Thank you, we corrected that.

10. Page 9, line 1, 2: should be "ship emissions". Supplement, page 9, line 24 and 25: Thank you, we corrected them.

11. Page 9, line 13: the grey filters from 2007, 2008: which study was this? Was it mentioned in the study? Please cite. Supplement, page 10, line 4: It was personal communication with Dr. Ofelia Rempillo.

12. Page 9, line 17: A re-definition of LTR. Supplement, page 10, line 10: Thank you, we corrected that.

13. Page 10, line 9: Please remove ":" Supplement, page 11, line 14: Thank you, we corrected that.

14. Table 1: This table display and format could be modified so that it is easier to pick out important information. There are too many brackets, e.g. Average sulfate (stdev) (ng/m3), hence confusing. Also the authors should avoid using too many horizontal and vertical lines in the table. Supplement, page 20, Table 2: The format of the table has been changed.

15. Figure 2: The time duration of the 3 graphs are not the same. I suggest that the time duration should match the sampling interval (8-24 July?), and please specify when support data is not available. Supplement, page 23, Figure 2: The time duration has been changed for the sampling period and more information has been added in the caption.

16. Figure 3: I would suggest using different color codes for SS and NSS. Also, it seems that in Figure 3a, sea salt sulfate was higher than total sulfate (second point from the top). It would be good to have detailed temporal data in number, so that it is easier to use and compare later, not just as average as currently in Table 1. Supplement, page 24, Figure 3: The color codes for SS and NSS have been changed. Also, concentration values have been added to the figure.

17. Figure 5: This figure is blurry and hard to read. Also, it should be $SO_4^{2-}$. Please also specify which day/which samples were considered more "Arctic", as it is difficult to flip back and forth to the transect figure to find out. I would suggest to name the sample 1, 2, 3, 4, 5, 6 or something, and keep the same consistent names in relevant figures and discussions. Supplement, page 27, Figure 6: Sampling intervals have been added to the figure. Also, Arctic and sub-Arctic samples have been distinguished in the caption of figure.

Please also note the supplement to this comment:
https://acp.copernicus.org/preprints/acp-2015-1010/acp-2015-1010-AC1-supplement.pdf

**Supplement:**

[revised manuscript text omitted]
 10th to 9:00 h on July 11th, and 14:00 h on July 15th to 10:35 h on July 17th. Wind speed and tempreatures were nor recorded before July 11th.

[Figure]

Figure 3. TOTAL sulfate, sea salt (a) and non-sea salt (b) sulfate concentrations (ng/m$^3$) of aerosols on A$_{>7.2\ \mu m}$-F$_{<0.49\ \mu m}$ filters. Numbers in the figure show TOTAL, sea salt and non-sea salt sulfate concentrations (ng/m$^3$) in gray, blue, and red colors respectively.

[Figure]

Figure 4. Total $\delta^{34}$S versus the percentage of sea salt sulfate of size fractionated aerosols. The mixing lines show sea salt/biogenic sulfate (solid line) and sea salt/anthropogenic sulfate (dashed line)

contributions. The standard deviations of each run were taken as the uncertainty for $\delta^{34}$S values.

[Figure]

2 Figure 5. FLEXPART-WRF backward configuration of potential emission sensitivity plots for a) July

3 13 (12:01:00 h), July 20 (12:17:00 h), and July 21 (12:01:00 h). The black line shows the ship track

4 (note that these panels include the ship track after July 23th 2014 when high volume sampling was not

5 performed). The airmass residence time (seconds) before arriving at the ship location is shown with

6 different colors. Numbers on the panels show the approximate lifetime and the center of the plume

7 locations.

[Figure]

Figure 6. Non-sea salt (a), anthropogenic (b) and biogenic (c) of non-sea salt sulfate concentrations, for size segregated aerosols in the Arctic and sub-Arctic. Strictly Arctic samples include thoes collected after July 13[th]. Inserts contain the first sampling period (9-11 July) in the Gulf of St. Lawrence.

[Figure]

Figure 7. The isotope ratio ($\delta^{34}S$ value) for $F_{<0.49 \mu m}$ and $SO_2$ filters along with the 1:1 line. Two
samples with different $\delta^{34}S$ values for $SO_2$ and $F_{<0.49 \mu m}$ filter sulfate are shown with asterisks.

---

## Author Comment (AC2) · 31 Mar 2016

Dear Dr. Alexander,

We are very thankful for your great comments. We believe that we have addressed all of the concerns. Please find the attached supplement: the revised sentences and sections in the supplement are highlighted with green color.

Yours Sincerely,

Roghayeh Ghahremaninezhad, (PhD Candidate) Department of Physics and Astronomy, University of Calgary, Tell: +1 403 708 2332 Email address: r.gh.phy@gmail.com rghahrem@ucalgary.ca Comments and answers: 1. Since they use Na+ to correct for sea-salt sulfate, and sea salt sulfate is such a large fraction of total sulfate, this is critical as it strongly influences the value of their calculated $\delta$34S nss and thus their conclusions. Supplement, page 6, line 3: The concentration of Na+ in blank papers has been reported.

2. Page 6: What is the analytical precision of the S-isotope measurements and how was it determined? Supplement, page 5, line 30: The standard deviation in replicate measurements of the standards is $\pm$0.3 ‰

3. Page 7: Wind speed also influences DMS emissions. Supplement, page 6, line 27: The "wind speed" has been added to the sentence.

4. Page 7 and throughout the paper: There are a lot of seemingly quantitative statements in the manuscript without the numbers in the text to back them up. For example, on page 7 line 13, how much less sea salt sulfate does it contain? Page 8 line 8, "the majority of sulfate" – what fraction is "majority"? Page 9 line 10, define what you mean by "high" and "low". Page 9 line 22: what percent makes this "important"? Page 9 line 23, what percent makes this "dominant"? Supplement, page 7, line 8; page 7, line 26; page 8, lines 25 and 26; page 9, line 8; page 9, line 10: Fractions have been added.

5. Page 8 line 17: Also should cite Jaeglé et al. [2011]. Supplement, page 8, line 7: Thank you, we referred to this paper.

6. Page 10: The FLEXPART-WRF model results should be presented. I was expecting to see a plot of the back trajectories but this seems to be missing. Supplement, page 4, line 16; page 9, line 16; page 25, Figure 5: Some examples of FLEXPART-WRF results have been added (Fig 5).

7. Page 10 line 2: There is no evidence from the isotope data for a significant contribution: Supplement, page 9, line 14: Thank you, we corrected that.

8. Page 10 line 19: How were sensitivity tests performed? Are you running a model? More detail is needed here. Supplement, page 9, lines 25 and 26: We did not run a model. "Sensitivity test" has been changed to "analysis" to make this clearer.

9. Page 11 line 20: Again, plots showing the results of the back trajectory calculations would be useful to show in a figure and referred to here. Supplement, page 10, line 20: Some examples of FLEXPART-WRF results have been added (Fig 5).

10. Paragraph beginning on Page 11 Line 21: Shouldn't you be discussing the biogenic contribution here? It seems weird to ignore it here when it's so important. Supplement, page 11, line 3: More information about biogenic contributions has been added.

11. Page 12 line 6: replace "in solution" with "the aqueous phase". Supplement, page 11, line 15: Thank you, we corrected that.

12. Page 12 lines 6-8: Cloud pH also strongly influences the rate of aqueous-phase reactions. Supplement, page 11, line 18: Thank you, we corrected that.

13. Page 12 lines 14-19: How were these numbers calculated? You have to assume some value for $\delta$34S (SO2) which is not stated here. Supplement, page 6, line 19: We used the isotope values from other studies. We added a sentence to make this clear.

14. Page 13 line 4: Would this make these samples biased high in the calculated anthropogenic fraction? Supplement, page 12, line 6: No, we compared $\delta$34S (SO2) with $\delta$34S (SO42-) (Figure 7). Results show that two samples (collected on July 15-17 and 17-19) contained more aerosols from anthropogenic sources (Table 3: $\sim$ 75% and $\sim$60% from anthropogenic sources). However for these samples the dominant source of SO2 was biogenic (80% of SO2 was from biogenic sources).

15. Page 13 line 22: from 0.49 to 0.95. Supplement, page 12, line 29: Thank you, we corrected that.

16. Table 1: Include fraction of nss-SO4 here. Supplement, page 20, Table 2: The fraction has been added.

17. Figure 4: What do the error bars represent in Figure 4 and how were they calculated? There is no dashed line in my version of the figure. Supplement, page 25, Figure 4, caption: The standard deviations of each run were taken as the uncertainty for ïĄd'34S values.

18. Figure 5: I would find this figure more useful if b and c showed fractions instead of absolute concentrations. Supplement, page 11, line 11 and page 21, Table 3: Table 3 has been added to report fraction of biogenic sources for each size range.

Please also note the supplement to this comment:
https://acp.copernicus.org/preprints/acp-2015-1010/acp-2015-1010-AC2-supplement.pdf

**Supplement:**

[revised manuscript text omitted]

---

## Author Response (AR1)

Dear Dr. Ervens,

Please find attached the Revised Version of the manuscript acp-2015-1010, "Title: Biogenic, anthropogenic, and sea salt sulfate size-segregated aerosols in the Arctic summer".

We are very thankful for the insightful comments of the Reviewers. We believe that we have addressed all of the concerns raised by Reviewers. Please find the attached responses to the comments in separate replies for reviewers:   the revised sentences and sections in the manuscript are highlighted with yellow and green colors in response to Reviewers comments.

Best Regards,

Roghayeh Ghahremaninezhad, (PhD Candidate)
Department of Physics and Astronomy,
University of Calgary,

Tell: +1 403 708 2332

Email address: r.gh.phy@gmail.com rghahrem@ucalgary.ca

**Subject:** Reply to the Interactive comment of Reviewer # 1 on "Biogenic, anthropogenic, and sea salt sulfate size-segregated aerosols in the Arctic summer"

    (The revised sentences and sections in the supplement are highlighted with yellow color)

Comments and answers:

1.

Page 2, line 23: "Sea salt enters the atmosphere via mechanical processes such as sea spray and bubble bursting" – this sentence is ambiguous. It could be good to explain briefly how sea spray aerosol is formed, with relevant references, such as (Lewis and Schwartz, 2004; Quinn et al., 2015).
Supplement, page 2, line 25 to page 3 line 3:    More details were added and we have now referred to "Lewis and Schwartz, 2004" and "Quinn et al., 2015".

2.
Page 4, line 12-14: "The high volume sampler was turned off manually to avoid contamination when the ship's emissions toward the sampler were observed or at times when the ship was stationary" - Can you specify how often / how long are these period?
Supplement, page 4, line 27:    Table 1 (page 19) has been added to show periods when the high volume sampler was off for more than 30 minutes and why it was turned off.

3.

Section 2: What are the uncertainties of the CF-IRMS?

Supplement, page 5, line 30:    The standard deviation in replicate measurements of the standards is ±0.3 ‰.

4.

Section 2: Please comment on the performance / uncertainty of the cascade impactor and how they might affect your results.

Supplement, page 4, line 29:   More details were added on the performance / uncertainty of the cascade impactor.

5.

Section 2: So how many samples did you collect in total? If the sampling period is 16 days (8-24 July) and your sampling interval is 2 days, then did you have 8 samples? Then why in Figure 3 you seemed to have only 6 data points? Please explain.

Supplement, page 4, line 28, and caption of figure 1 (page 22):   Sampling intervals for the high volume sampler:  from 9 to 22 July (9-11, 11-13, 13-15, 15-17, 17-19, 20-22). Six samples were collected. The first day (July 8) was close to Quebec City so sampling was started the next day. Also, the high volume sampler was off because of stormy weather from 10:00 h on July 19[th] to 10:00 h on July 20[th].

6.

Section 2, page 5, line 21-24: You cited some sulfur isotope apportionment in the Arctic. Did you use this in your calculations shown here? Please specify.

Supplement, page 6, line 19:   Yes, we used the values. We added a sentence to make this clear.

7.

Section 2: Please include some short description of $\delta^{34}S$.

Supplement, page 5, line 26:   Description of $\delta^{34}S$ has been added.

8.

Page 8, line 15: "shows" should be "show".

Supplement, page 9, line 15:   Thank you, we corrected that.

9.

Page 8, line 17: Please remove ":".

Supplement, page 9, line 17:   Thank you, we corrected that.

10.

Page 9, line 1, 2: should be "ship emissions".

Supplement, page 9, line 24 and 25:   Thank you, we corrected them.

11.

Page 9, line 13: the grey filters from 2007, 2008: which study was this? Was it mentioned in the study? Please cite.

Supplement, page 10, line 4:   It was personal communication with Dr. Ofelia Rempillo.

12.

Page 9, line 17: A re-definition of LTR.

Supplement, page 10, line 10:    Thank you, we corrected that.

13.

Page 10, line 9: Please remove ":"

Supplement, page 11, line 14:    Thank you, we corrected that.

14.

Table 1: This table display and format could be modified so that it is easier to pick out important information. There are too many brackets, e.g. Average sulfate (stdev) (ng/m3), hence confusing. Also the authors should avoid using too many horizontal and vertical lines in the table.

Supplement, page 20, Table 2:   The format of the table has been changed.

15.

Figure 2: The time duration of the 3 graphs are not the same. I suggest that the time duration should match the sampling interval (8-24 July?), and please specify when support data is not available.

Supplement, page 23, Figure 2:    The time duration has been changed for the sampling period and more information has been added in the caption.

16.

Figure 3: I would suggest using different color codes for SS and NSS. Also, it seems that in Figure 3a, sea salt sulfate was higher than total sulfate (second point from the top). It would be good to have detailed temporal data in number, so that it is easier to use and compare later, not just as average as currently in Table 1.

Supplement, page 24, Figure 3:    The color codes for SS and NSS have been changed. Also, concentration values have been added to the figure.

17.

Figure 5: This figure is blurry and hard to read. Also, it should be $SO_4^{2-}$. Please also specify which day/which samples were considered more "Arctic", as it is difficult to flip back and forth to the transect figure to find out. I would suggest to name the sample 1, 2, 3, 4, 5, 6 or something, and keep the same consistent names in relevant figures and discussions.

Supplement, page 27, Figure 6:   Sampling intervals have been added to the figure. Also, Arctic and sub-Arctic samples have been distinguished in the caption of figure.

**Subject:** Reply to the Interactive comment of Dr. Alexander (Reviewer # 2) on "Biogenic, anthropogenic, and sea salt sulfate size-segregated aerosols in the Arctic summer"

(The revised sentences and sections in the supplement are highlighted with green color)

Comments and answers:

1.

Since they use Na+ to correct for sea-salt sulfate, and sea salt sulfate is such a large fraction of total sulfate, this is critical as it strongly influences the value of their calculated $\delta^{34}S$ nss and thus their conclusions.

Supplement, page 6, line 3: The concentration of $Na^+$ in blank papers has been reported.

2.

Page 6: What is the analytical precision of the S-isotope measurements and how was it determined?

Supplement, page 5, line 30: The standard deviation in replicate measurements of the standards is ±0.3 ‰.

3.

Page 7: Wind speed also influences DMS emissions.

Supplement, page 6, line 27: The "wind speed" has been added to the sentence.

4.

Page 7 and throughout the paper: There are a lot of seemingly quantitative statements in the manuscript without the numbers in the text to back them up. For example, on page 7 line 13, how much less sea salt sulfate does it contain? Page 8 line 8, "the majority of sulfate" – what fraction is "majority"? Page 9 line 10, define what you mean by "high" and "low". Page 9 line 22: what percent makes this "important"? Page 9 line 23, what percent makes this "dominant"?

Supplement, page 7, line 8; page 7, line 26; page 8, lines 25 and 26; page 9, line 8; page 9, line 10: Fractions have been added.

5.

Page 8 line 17: Also should cite Jaeglé et al. [2011].

Supplement, page 8, line 7:   Thank you, we referred to this paper.

6.

Page 10: The FLEXPART-WRF model results should be presented. I was expecting to see a plot of the back trajectories but this seems to be missing.

Supplement, page 4, line 16; page 9, line 16; page 25, Figure 5:    Some examples of FLEXPART-WRF results have been added (Fig 5).

7.

Page 10 line 2: There is no evidence from the isotope data for a significant contribution:

Supplement, page 9, line 14:   Thank you, we corrected that.

8.

Page 10 line 19: How were sensitivity tests performed? Are you running a model?

More detail is needed here.

Supplement, page 9, lines 25 and 26:    We did not run a model.  "Sensitivity test" has been changed to "analysis" to make this clearer.

9.

Page 11 line 20: Again, plots showing the results of the back trajectory calculations would be useful to show in a figure and referred to here.

Supplement, page 10, line 20:    Some examples of FLEXPART-WRF results have been added (Fig 5).

10.

Paragraph beginning on Page 11 Line 21: Shouldn't you be discussing the biogenic contribution here? It seems weird to ignore it here when it's so important.

Supplement, page 11, line 3:   More information about biogenic contributions has been added.

11.

Page 12 line 6: replace "in solution" with "the aqueous phase".

Supplement, page 11, line 15:   Thank you, we corrected that.

12.

Page 12 lines 6-8: Cloud pH also strongly influences the rate of aqueous-phase reactions.

Supplement, page 11, line 18:   Thank you, we corrected that.

13.

Page 12 lines 14-19: How were these numbers calculated? You have to assume some value for $\delta^{34}S$ ($SO_2$) which is not stated here.

Supplement, page 6, line 19:   We used the isotope values from other studies. We added a sentence to make this clear.

14.

Page 13 line 4: Would this make these samples biased high in the calculated anthropogenic fraction?

Supplement, page 12, line 6: No, we compared $\delta^{34}S$ ($SO_2$) with $\delta^{34}S$ ($SO_4^{2-}$) (Figure 7). Results show that two samples (collected on July 15-17 and 17-19) contained more aerosols from anthropogenic sources (Table 3: ~ 75% and ~60% from anthropogenic sources). However, for these samples the dominant source of $SO_2$ was biogenic (80% of $SO_2$ was from biogenic sources).

15.

Page 13 line 22: from 0.49 to 0.95.

Supplement, page 12, line 29:   Thank you, we corrected that.

16.

Table 1: Include fraction of nss-SO4 here.

 Supplement, page 20, Table 2:   The fraction has been added.

17.

Figure 4: What do the error bars represent in Figure 4 and how were they calculated?

There is no dashed line in my version of the figure.

Supplement, page 25, Figure 4, caption:   The standard deviations of each run were taken as the uncertainty for $\delta^{34}S$ values.

18.

Figure 5: I would find this figure more useful if b and c showed fractions instead of absolute concentrations.

Supplement, page 11, line 11 and page 21, Table 3:   Table 3 has been added to report fraction of biogenic sources for each size range.

[revised manuscript text omitted]